# Unleashing the Power of 2D Diffusion Representation for High Fidelity 3D Generation

## Abstract

State-of-the-art (SOTA) approaches for 3D content generation are predominantly built upon a sequential framework: first generating geometric shapes, followed by texture estimation that leverages geometric cues. Consequently, these methods typically incur computational costs on the order of minutes when generating both geometry and texture, and often suffer from significant shape-texture misalignment—a limitation attributed to the sequential decoupling of these two stages. To mitigate these limitations, recent works have aimed to jointly model geometry and texture within a unified framework, which in turn enhances shape-texture consistency. Nevertheless, these joint approaches still face challenges in precise texture modeling, largely due to the loss of fine-grained texture details during latent feature learning. To address this remaining challenge, in this work, we propose a novel joint architecture that not only preserves the advantage of unifying geometry and texture modeling but also retains and effectively captures fine-grained texture details by integrating image diffusion features into the latent feature learning process. We further recognize that modeling such fine-grained texture features presents notable challenges, which arise from the inherent complexity of mapping 2D visual details onto 3D surfaces. To alleviate this challenge, we introduce a diffusion-based module that enhances cross-modal alignment between 3D structures and 2D image inputs, thereby enabling the direct learning of rich, fine-grained texture features from 2D image conditions. Extensive empirical evaluations demonstrate that our approach results in a 3D content generation algorithm that outperforms existing SOTA approaches, delivering substantial improvements in texture modeling quality.

## 1 Introduction

Image-to-3D generation has emerged as a pivotal subfield within the broader domain of 3D AI-Generated Content (AIGC), whose core task is centered on synthesizing high-fidelity 3D assets from a single 2D image of an arbitrary object. Recent years have witnessed remarkable advancements in large-scale 3D generative models, driving significant progress in synthetic 3D content creation. A dominant paradigm among these approaches employs a sequential framework Zhang et al. (2024b); Li et al. (2025b); Zhang et al. (2023); Li et al. (2025a): first, geometric structures are generated via a diffusion model operating over a compact latent space, with the latent representation encoded by a pretrained variational autoencoder (VAE); second, texture details are synthesized using a multiview image generation model, which leverages geometric cues from the precomputed shape. However, this sequential design inherently introduces substantial inefficiencies: (1) the sequential generation of shape and texture occurs across two distinct, computationally expensive modules; (2) reliance on large-scale multiview diffusion models often results in generation times on the order of minutes per 3D object. A second critical limitation is the inherent misalignment between geometry and texture: since shape and texture are modeled separately without explicit cross-modal constraints, the synthesized texture frequently fails to map accurately to the underlying geometry, degrading the overall fidelity of the generated 3D asset.

Beyond the sequential paradigms outlined above, a parallel line of research has investigated models that jointly generate 3D geometry and corresponding texture Tang et al. (2024); Xiang et al. (2025). A subset of these approaches leverages large-scale transformer architectures to directly learn 3D Gaussian Splatting (3DGS) Tang et al. (2024) in an end-to-end manner; however, the resulting ge-

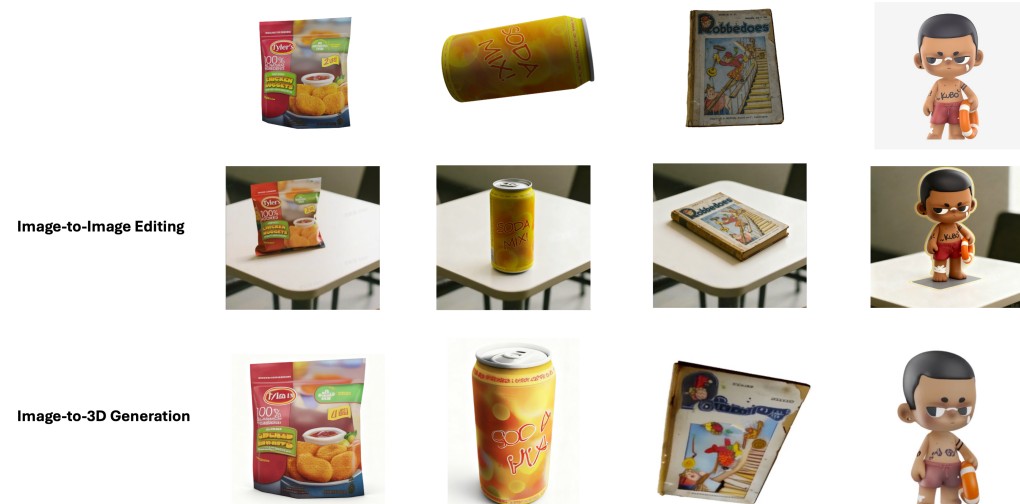

Figure 1: Texture-focused comparison between SOTA image-to-image(I2I) editing model and image-to-3d(I3D) model. Specifically, I2I editing models demonstrate superior performance in preserving strict visual coherence with the conditional input images, whereas I3D models struggle to achieve this level of coherence.

ometries tend to be excessively smooth and lacking fine-grained structural details, while the accompanying textures often suffer from noticeable blurriness. Recent advances Xiang et al. (2025) have shifted toward leveraging sparse 3D structures for compact 3D object representation. These methods typically employ 3D binary occupancy masks to delineate the valid spatial regions of the target object, thereby significantly reducing computational and memory overhead relative to dense 3D representations. Nevertheless, two critical bottlenecks persist: First, the DINOv2 features adopted in such approaches are typically designed for coarse-grained semantic representation, rendering them incapable of capturing fine-grained texture details—an essential requirement for preserving input image fidelity in 3D generation. Second, sparse 3D structures and 2D images are inherently misaligned in both their representation modalities and dimensionalities. The naive application of attention mechanisms—whose primary function is to measure feature similarity within homogeneous modalities (e.g., 2D-to-2D, 3D-to-3D)—fails to establish meaningful correlations between heterogeneous 2D and 3D data. This undermines the model's capacity to faithfully preserve fine-grained visual details from input 2D image conditions. Such structural and dimensional mismatch poses an insurmountable barrier to effective cross-modal feature alignment, ultimately limiting the transfer of high-fidelity visual information to the generated 3D asset. As illustrated in fig. 1, state-of-the-art 2D image-to-image editing models Batifol et al. (2025); Wang et al. (2025) excel at maintaining strict visual coherence with the conditional input image: they faithfully preserve both low-level fine-grained textural details and high-level semantic consistency. In contrast, existing joint 3D generation methods Xiang et al. (2025) struggle to retain this level of consistency, frequently producing textures that are blurry or devoid of input-specific fine details. This discrepancy underscores a critical gap in visual detail preservation between mature 2D conditional modeling paradigms and emerging joint 3D generation frameworks.

In this work, we propose Diff2to3 to address the aforementioned challenges in joint image-to-3D generation. For 3D asset representation, we adopt 3D Gaussian Splatting (3DGS) Kerbl et al. (2023)—a representation that provides two key advantages: high-fidelity appearance preservation and efficient volumetric rendering. Our approach builds upon the joint framework proposed in Xiang et al. (2025) and introduces targeted architectural and algorithmic enhancements to both the 3D latent reconstruction and 3D content generation pipelines. In the latent learning stage, we modify the input feature composition of the original VAE architecture. Specifically, we leverage rich, prelearned visual features from SOTA pretrained large-scale image diffusion models Labs (2024). These models are trained on large-scale natural image datasets, enabling them to encode a comprehensive understanding of both low-level reconstructive cues and high-level semantic relation-

ships. In contrast to prior works that utilize DINOv2 features Oquab et al. (2023)—which prioritize high-level scene semantics and are less optimized for capturing fine-grained details. Our use of image diffusion features alleviates the inherent information loss during latent encoding, which ensures the learned latent representation retains the fine-grained texture details and subtle geometric hints necessary for maintaining cross-modal consistency between the input 2D image and the generated 3D asset. In the generation phase, we observe that fine-grained textural features from image diffusion model—characterized by subtle chromatic variations, intricate surface patterns, and microscale details—remain notoriously challenging to model, especially under the constraints of an efficient sparse-structured latent space. Empirical observations indicate that research explicitly focused on modeling such fine-grained features in this constrained setting remains relatively limited; furthermore, the naive adoption of diffusion architectures optimized for coarse-grained texture modeling fails to capture these fine-grained characteristics with adequate efficiency. To address this critical gap, we draw inspiration from SOTA 2D diffusion paradigms Labs (2024) and introduce two novel components specifically designed for the sparse 3D structural scenario: (1) the Sparse-structure Multi-modal Diffusion Transformer (SMDiT), a diffusion-based transformer architecture that jointly models the probability distributions of sparse voxel-based latents and conditional 2D image tokens. This design enables cross-modal interaction and knowledge transfer, thereby facilitating the propagation of fine-grained visual cues from 2D inputs to 3D representations; (2) Modal-Aware Rotary Position Embedding (MARoPE), which explicitly encodes the intrinsic spatial relationships between 3D voxel coordinates (defined in world space) and 2D image pixels (localized in the input image plane). By quantifying these geometric correspondences, MARoPE enables implicit alignment between local 3D regions and their corresponding visual details in the 2D input, thus enhancing the fidelity of texture transfer. Through the integration of these proposed enhancements, our model achieves more efficient learning and faithful preservation of fine-grained textural features, thereby addressing the limitations of prior approaches in capturing high-resolution visual details within sparse 3D latent spaces.

The key contributions of our work are summarized as follows:

- We leverage prelearned visual features from large-scale image diffusion models to construct input 3D feature volumes for the VAE, enabling the learning of a more expressive latent space that captures both low-level reconstructive cues and high-level semantics. This leads to improved reconstruction accuracy relative to prior approaches.

- We propose the Sparse-structure Multi-modal Diffusion Transformer (SMDiT)—a novel architecture specifically designed for modeling sparse-structured 3D latent spaces—and Modal-Aware Rotary Position Embedding (MARoPE), which collectively enhance cross-modal alignment between 3D structures and 2D image conditions.

- Extensive quantitative and qualitative experiments on standard image-to-3D benchmarks show that our efficient joint approach generates 3D assets with high-fidelity, image-consistent textures, outperforming SOTA methods across key evaluation metrics.

## 2 PRELIMINARIES

**Sparse Voxel-based Representation.** Introduced by Xiang et al. (2025), the sparse voxel-based representation constitutes a unified 3D latent framework that encodes 3D objects by distributing latent features across a sparsely populated voxel grid. This design avoids redundant storage of non-informative (i.e., empty or irrelevant) voxels, thereby striking a balance between computational efficiency and representational fidelity. Formally, a 3D asset is represented as a set of tuples $\{(z_i, p_i)\}_{i=1}^L$, where $p_i \in \{0, ..., N-1\}$ denotes the positional index of an active (i.e., non-empty) voxel within a cubic grid of resolution $N$, and $z_i \in \mathbb{R}^C$ is a local latent vector associated with $p_i$. Given that the number of active voxels is substantially smaller than the total grid size ($L \ll N^3$), this framework enables efficient high-resolution modeling while preserving the precise spatial locality of both geometric structure and appearance attributes. Furthermore, it supports flexible decoding into diverse 3D representations (e.g., 3D Gaussian Splatting, NeRF, or mesh) via task-specific decoder heads, enhancing its adaptability across downstream applications.

The generation of such sparse voxel-based representations typically adheres to a two-stage pipeline. In the first stage, a Diffusion Transformer (DiT) Peebles & Xie (2023) is trained to generate the coordinates of the feature grid $\{p_i\}_{i=1}^L$ from a noise distribution, effectively modeling the spatial

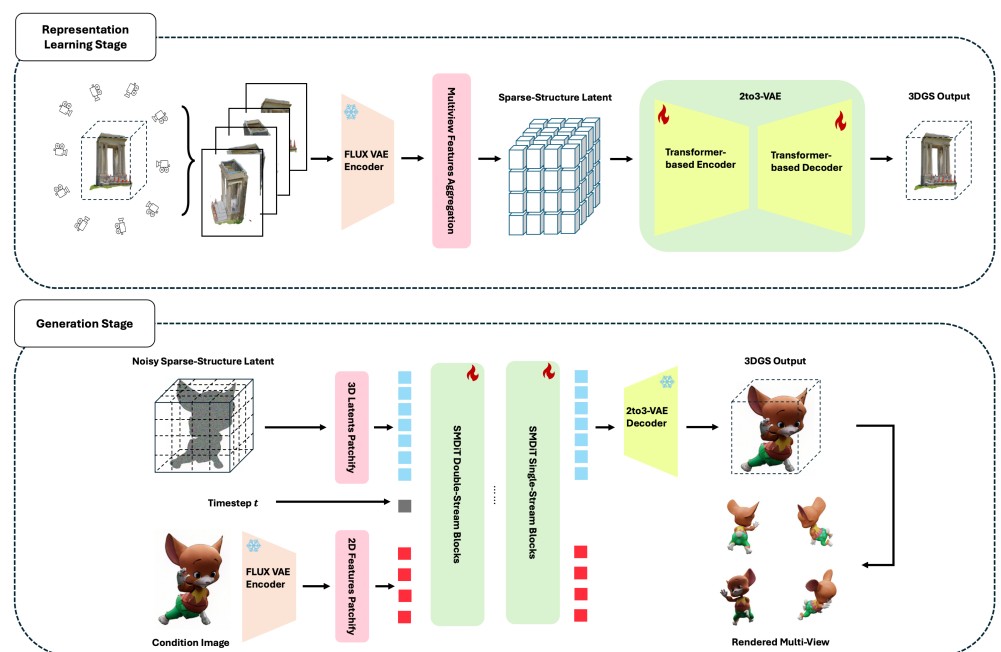

Figure 2: The overall framework of Diff2to3 is visually illustrated as follows: During the representation learning stage, FLUX features of multi-view images are extracted to construct sparse-structured latent representations, which serve to support 3D latent reconstruction. In the generation stage, these learned latent representations are generated via the SMDiT module under the condition of a given conditional image. The learning process is further enhanced using the same sparse voxel-based technique proposed in Xiang et al. (2025); for simplicity, this technique is omitted from the figure.

arrangement of active voxels. In the second stage, an additional DiT—conditioned on the precomputed coordinates $\{p_i\}_{i=1}^{L}$ is employed to generate the corresponding latent features $\{z_i\}_{i=1}^{L}$, which encode geometric and textural information.

**3D Gaussian Splatting.** 3D Gaussian Splatting (3DGS) Kerbl et al. (2023) constitutes a real-time radiance field representation that models 3D scenes or objects via a collection of discrete 3D Gaussian primitives, where each primitive functions as a compact unit for encoding local geometric structure and appearance attributes. Formally, each 3D Gaussian is defined by a parameter set $\Theta = \{\mathbf{x}, \mathbf{s}, \mathbf{q}, \alpha, \mathbf{c}\}$, where $\mathbf{x} \in \mathbb{R}^3$ denotes the center coordinate, anchoring its spatial position; $\mathbf{s} \in \mathbb{R}^3$ is a scaling vector that adjusts the primitive's dimensions along orthogonal axes; and $\mathbf{q} \in \mathbb{R}^4$ represents a rotation quaternion, enabling orientation adjustments to fit complex surface geometries. For photorealistic rendering, each Gaussian further includes an opacity value $\alpha \in \mathbb{R}$ and color information $\mathbf{c} \in \mathbb{R}^d$ - typically represented via spherical harmonics to model view-dependent effects such as specularity or shading variations. The 3DGS rendering pipeline operates by first projecting each 3D Gaussian onto the 2D image plane, yielding a perspective-aligned 2D Gaussian. Subsequently, front-to-back per-pixel alpha compositing is applied to blend the color and opacity of overlapping Gaussians, producing the final pixel values. This approach circumvents the time-consuming ray marching inherent to NeRF-based methods and the topological constraints of mesh representations, thereby enabling real-time photorealistic rendering— a key advantage for interactive applications and high-fidelity visualization tasks.

## 3 METHOD

In this section, we detail the architectural design of the Diff2To3 framework, which is tailored to generate 3DGS representations of objects conditioned on a single input image. As illustrated in fig. 2, the framework consists of two core stages: representation learning and 3DGS generation. In section 3.1, we first analyze the latent encoding scheme proposed in prior work, identify its inher-

ent limitations, and then present our revised input volume construction strategy—this modification enables the learning of a more efficient and expressive latent space, specifically optimized for detailed 3DGS representation. Next, in section 3.2, we elaborate on the proposed flow-based model architecture, alongside a novel position embedding design tailored explicitly to the sparse-structured latent space; this embedding design facilitates effective modeling of cross-modal dependencies between 2D image cues and 3D structural features. Finally, in section 3.3, we introduce the training objectives adopted to optimize the framework.

## 3.1 2TO3 VARIATIONAL AUTOENCODER

Our 2to3-VAE is constructed based on the VAE architecture proposed in Xiang et al. (2025), with targeted modifications motivated by a key observation: the granularity of input features directly influences the quality and expressiveness of the latent representations learned by the VAE. In Xiang et al. (2025), each 3D asset is first voxelized into a feature volume; the feature of each active voxel is derived by aggregating DINOv2 features Oquab et al. (2023) extracted from densely rendered multi-view images of the asset. While DINOv2 performs well in general-purpose representation learning, its design prioritizes high-level semantic abstraction—often at the cost of pixel-level appearance details that are critical for achieving faithful image-to-3D alignment. This misalignment with the core requirement of the image-to-3D task (preserving fine-grained visual consistency) constitutes a major limitation. A second critical limitation of the original encoding scheme stems from its aggressive dimensionality compression: DINOv2's 1024-dimensional features are compressed into 8-dimensional latents, resulting in a compression ratio of 128:1. This extreme downsampling induces significant loss of fine-grained information, further impairing the model's ability to preserve texture details and compromising the fidelity of 3D outputs.

To address these limitations, we revise the input feature pipeline of 2to3-VAE by utilizing FLUX features Labs (2024) as the input to the encoder. Unlike DINOv2, FLUX is explicitly optimized to preserve low-level visual details and reconstructive cues—properties that are well-aligned with the requirement of the image-to-3D task for precise 2D-to-3D visual consistency. FLUX has already demonstrated superior performance in high-fidelity image reconstruction tasks, making it well-suited for extension to 3D generation scenarios where retaining textural nuances is critical. These features inherently encode precise visual details (e.g., color gradients, surface patterns) that bridge 2D input images and 3D textures, thereby enhancing cross-modal alignment. Furthermore, FLUX features exhibit a more favorable dimensionality profile: they are compact yet information-dense 16-dimensional vectors. When compressed to the target 8-dimensional latents, this dimensionality reduction ($16 \rightarrow 8$) results in a modest compression ratio of 2:1—far lower than DINOv2's 128:1. This reduced compression ratio minimizes information loss, enabling the retention of fine-grained textural details in the latent space. By leveraging FLUX's reconstructive focus and efficient dimensionality, we align the input feature pipeline of 2to3-VAE more effectively with the demands of image-to-3D generation, laying a robust foundation for high-fidelity 3DGS outputs.

## 3.2 SPARSE-STRUCTURE MULTI-MODAL DIFFUSION TRANSFORMER

**Network Architecture.** Our SMDiT network design is motivated by the following observation: 2D and 3D features belong to distinct modalities; simply adopting the attention mechanism Xiang et al. (2025) is insufficient to capture the correspondences between these two modalities, thereby failing to generate detailed 3D objects that align well with input images. These two modalities therefore necessitate a pre-alignment module to ensure they can be well aligned in the latent space while retaining their unique characteristics. Inspired by image diffusion models Labs (2024), our network is built from a combination of double-stream and single-stream blocks, as illustrated in fig. 3. The double-stream blocks first process noisy sparse voxel tokens and conditioned image tokens separately using modality-specific weights, thus preserving the unique characteristics of each input type. After modality-specific processing, joint attention is then applied to the concatenated token sequence to initiate cross-modal interaction. To further enhance the exchange of mutual information between the 2D and 3D modalities, the concatenated tokens are treated as a unified sequence and fed into subsequent single-stream blocks—blocks that are equivalent to standard transformer blocks. To improve training efficiency, we patchify both the noisy sparse voxel latents as well as the condition image features. To improve training efficiency, we patchify both the noisy sparse voxel latents and the conditioned image features. The former is achieved by employing downsampling blocks with

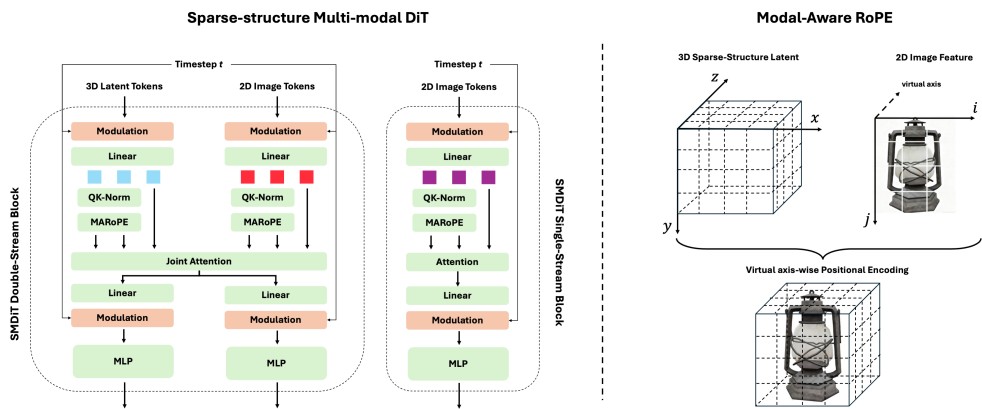

Figure 3: An illustration of our proposed Sparse-structured Multi-modal DiT (SMDiT) network architecture and Modal-Aware RoPE (MARoPE) strategy. SMDiT employs a transformer architecture comprising both double-stream and single-stream blocks. It incorporates the characteristics of the sparse-structured latent space and facilitates interactions between 3D and 2D modalities. (Note that green blocks contain learnable parameters, whereas orange blocks do not.) For the MARoPE strategy, 2D images are mapped onto a virtual plane outside the latent volume, while raw 3D voxel coordinates within the volume are retained.

sparse convolutions to group active voxels within a $2^3$ local region, while the latter is implemented by adopting the same $2^2$ patch partitioning as in the DiT design Esser et al. (2024), which aligns with standard image feature processing pipelines.

**Modal-Aware Rotary Position Embedding**. Previous works Su et al. (2024); Wu et al. (2025) have highlighted the importance of positional embeddings, as they help the model distinguish between modalities and learn spatial correspondences. However, we notice that current positional encoding strategies for cross-modal tasks are mostly designed for text-image or image-image pairs Wu et al. (2025); Batifol et al. (2025); Tan et al. (2024), and they fail to adequately address the unique challenges in image-to-3D for linking 2D image patches to 3D volumetric regions. A few works Feng et al. (2025) propose 3D-aware RoPE to explicitly encode 2D-3D correspondences, but it relies on accurate camera parameters to build a canonical coordinate map and provide this correspondence. This assumption is impractical for general-purpose 3D generation, where users rarely provide calibrated camera parameters, leading to geometric misalignment and degraded performance.

To address these limitations, we propose Modal-Aware Rotary Position Embedding (MARoPE), a novel positional encoding scheme that enables implicit cross-modal correspondence learning without relying on explicit geometric alignment. As shown in fig. 3, MARoPE explicitly distinguishes 2D and 3D modalities by mapping 2D image patch indices $(i, j)$ to a "virtual plane" at 3D coordinate $(i, j, z_{max} + 1)$ (outside the 3D volume's bounds) while retaining raw 3D voxel coordinates $(x, y, z)$ for the volume. This design conceptualizes the conditioned image as being attached along the third dimension of the latent volume. By embedding 2D and 3D tokens in a shared 3D space but with 2D tokens isolated on a virtual plane, MARoPE allows the diffusion model to learn implicit correspondences between 2D image patches and 3D volumetric regions. This ultimately enhances visual consistency without enforcing explicit geometric alignment.

### 3.3 TRAINING DETAILS

**VAE Training.** Our 2to3-VAE is trained in an end-to-end manner. At each step, we randomly choose one reference view, then minimize the perceptual L1 reconstruction loss between the rendering result of the VAE's decoded output and that of the ground-truth from the same viewing angle. Following Xiang et al. (2025), we also impose geometry regularizations for volume and opacity of the Gaussians, and KL constraints to push the latent space towards standard normal. The full training objective can be written as:

$$\mathcal{L}_{\text{total}} = \mathcal{L}_{\text{recon}} + \mathcal{L}_{\text{vol}} + \mathcal{L}_{\alpha} + \mathcal{L}_{KL} \tag{1}$$

| Method | SSIM↑ | PSNR↑ | LPIPS↓ |
|---|---|---|---|
| GaussianAnything | 0.9475 | 26.73 | 0.06397 |
| TRELLIS | 0.9712 | 30.14 | 0.02901 |
| Ours | **0.9773** | **31.86** | **0.02633** |

Table 1: Quantitative evaluation of 3D reconstruction with other latent representations in terms of appearance fidelity on Toys4k. **Bold** denotes the best results.

**Diffusion Training.** Our SMDiT model is trained with a flow-matching pipeline Lipman et al. (2022), a continuous normalizing flow framework that formulates generation as a progressive transformation of noise toward the target distribution. Specifically, it uses a linear interpolation forward process $z_t = (1-t)z_0 + t\epsilon, t \sim U(0,1), \epsilon \sim \mathcal{N}(\mathbf{0}, \mathbf{I})$ to derive noisy inputs $z_t$. Then, we parameterize the SMDiT model as $v_\theta$ to predict the velocity field $u_t$ of the noisy input $z_t$ with respect to the straight-line trajectory, using a conditional flow matching (CFM) objective:

$$\begin{aligned}
\mathcal{L}_{\text{CFM}}(\theta) &= \|v_t(z;\theta) - u_t(z|\epsilon)\|_2^2 \\
&= \|v_t((1-t)z_0 + t\epsilon;\theta) - (\epsilon - z_0)\|_2^2
\end{aligned} \tag{2}$$

## 4 EXPERIMENTS

In this section, we first present the implementation details, comparative baselines, and evaluation protocols. We then conduct two sets of experiments to evaluate the quantitative and qualitative performance of our method and other baseline methods across both reconstruction and generation tasks.

### 4.1 EXPERIMENT SETTINGS

**Datasets.** Following Xiang et al. (2025), we remove assets with low-quality textures and manually curate approximately 360K 3D assets from the 3D-FUTURE Fu et al. (2021), ABO Collins et al. (2022), HSSD Khanna et al. (2024), and Objaverse (XL) Deitke et al. (2023) datasets for model training. For quantitative and qualitative evaluation, we randomly sample 1,000 assets from the Toys4k dataset Stojanov et al. (2021).

**Implementation Details.** For reconstruction experiments, we render 150 images per asset; for diffusion-based generation experiments, we render 24 images per asset. These images are rendered with a set of different field-of-views (FoVs) and at a resolution of $512 \times 512$. During training, we adopt the AdamW optimizer Loshchilov & Hutter (2017) with a learning rate of $1 \times 10^{-4}$. During inference, we set the number of sampling steps to 50 to ensure fair comparison across all methods. All experiments are conducted on 8 NVIDIA A100 GPUs.

### 4.2 RECONSTRUCTION EXPERIMENTS

We first evaluate the appearance reconstruction fidelity of our 2to3-VAE, whose performance establishes the upper bound for the generation quality of the framework. Following Xiang et al. (2025), we report the SSIM, PSNR, and LPIPS metrics by comparing the rendered reconstruction results with the ground truth. We primarily compare our method against two baselines: TRELLIS Xiang et al. (2025) and GaussianAnything Lan et al. (2024). The former serves as the example of sparse structure 3D generation without 2D diffusion features, while the latter is an interactive latent space with a point cloud structure that is also trained on large-scale data. The commonly used vecset latent representation is employed for geometry reconstruction rather than texture modeling, and thus is not the focus of this work. As shown in table 1, our method outperforms all baselines across all evaluated metrics. This result validates the effectiveness of incorporating 2D diffusion features in constructing 3D feature volumes.

| Method | CLIP↑ | FD$_{incep}$↓ | KD$_{incep}$(%)↓ | FD$_{dinov2}$↓ | KD$_{dinov2}$(%)↓ | # Param. |
|--------|-------|---------------|------------------|----------------|-------------------|----------|
| Shap-E | 86.32 | 133.96 | 8.23 | 792.43 | 147.72 | 300 M |
| LGM | 87.23 | 65.89 | 1.33 | 662.91 | 75.49 | 415 M |
| InstantMesh | 89.95 | 45.38 | 2.54 | 392.15 | 24.48 | 866 M |
| GaussianAnything | 85.09 | 65.31 | 8.07 | 702.03 | 71.72 | 586 M |
| 3DTopia-XL | 86.10 | 64.39 | 1.16 | 619.11 | 64.28 | 909 M |
| TRELLIS | 98.02 | 24.57 | 0.0421 | 146.14 | 8.74 | 770 M |
| Ours | **98.45** | **19.78** | **0.0394** | **122.90** | **3.78** | 821 M |

Table 2: Quantitative evaluation of image-conditioned 3D generation on Toys4k dataset in terms of appearance fidelity of 2D renderings. **Bold** and underline respectively denote the best and the second-best results.

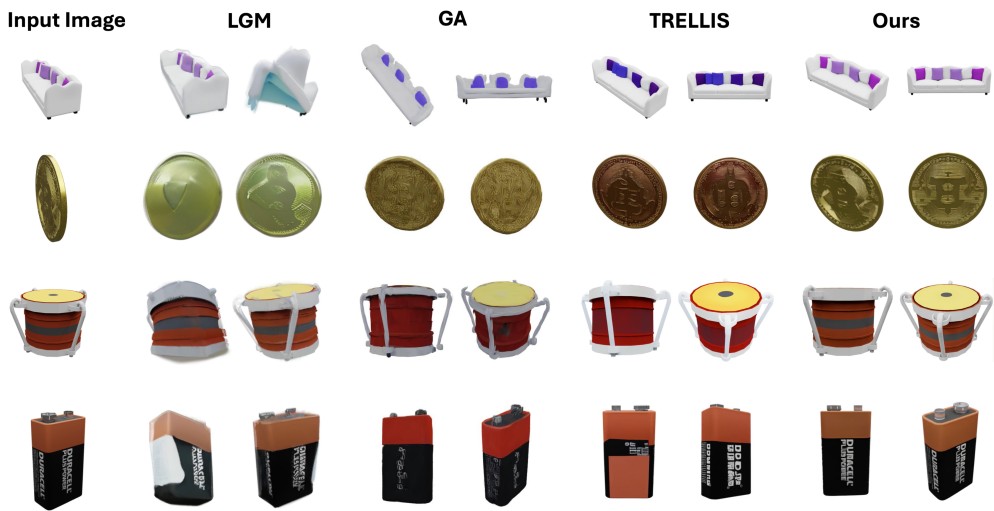

Figure 4: Visual comparisons with 3DGS-based generation method. *GA* is short for *GaussianAnything*. Please zoom in for clearer visualization.

### 4.3 GENERATION EXPERIMENTS

For generation quality, we conduct quantitative comparisons with SoTA 3D generation methods, including Shap-E Jun & Nichol (2023), LGM Tang et al. (2024), InstantMesh Xu et al. (2024a), GaussianAnything Lan et al. (2024), 3DTopia-XL Chen et al. (2025b), and TRELLIS Xiang et al. (2025)—all under the single-image input condition. Given our focus on appearance fidelity, we evaluate the visual consistency between 2D renderings of the generated 3D assets and the image prompts. To ensure the robustness of our evaluation, we employ a suite of metrics: Fréchet Distance (FD) and Kernel Distance (KD), each paired with distinct feature extractors, as well as the CLIP Score. As shown in table 2, our method outperforms previous approaches across all evaluated metrics.

We further present qualitative comparisons with 3DGS-based generation methods in fig. 4. As observed in the figure, LGM and GaussianAnything exhibit significant distortions in both shape and appearance, particularly when viewed from novel perspectives. TRELLIS, on the other hand, suffers from color misalignment and inconsistent textural details relative to the input image prompts. In contrast, our framework consistently produces results that best preserve the input's geometric structure, color accuracy, and textural details—such as the arrangement of sofa cushions, the surface features of coins, the stripe patterns on drums, and the branding on batteries—thereby demonstrating superior visual fidelity across all tested objects.

| Summary | Configurations | | | CLIP$\uparrow$ | FD$_{incep}\downarrow$ | KD$_{incep}(\%)\downarrow$ | FD$_{dinov2}\downarrow$ | KD$_{dinov2}(\%)\downarrow$ |
|---------|----------|-------|--------|------|------|------|------|------|
| | 2to3-VAE | SMDiT | MARoPE | | | | | |
| Exp 1 | ✗ | ✗ | ✗ | 98.02 | 24.57 | 0.0421 | 146.14 | 8.74 |
| Exp 2 | ✓ | ✗ | ✗ | 96.96 | 30.24 | 0.0523 | 210.50 | 5.58 |
| Exp 3 | ✓ | ✓ | ✗ | 97.11 | 28.74 | 0.0496 | 159.34 | 4.11 |
| Exp 4 | ✓ | ✓ | ✓ | **98.45** | **19.78** | **0.0394** | **122.90** | **3.78** |

Table 3: Ablation studies of image-conditioned 3D generation on different design components. **Bold** denotes the best results.

## 4.4 ABLATION STUDIES

In this section, we perform a series of ablation studies to systematically investigate the efficacy of the SMDiT architecture and MARoPE design. The experimental configurations are detailed in table 3. Specifically, Exp. 1 serves as our baseline, comprising (1) a DINOv2-based 3D VAE, (2) the DiT architecture, and (3) absolute position embeddings. We incrementally incorporate our proposed designs in subsequent experiments to validate their individual and combined effectiveness.

Our empirical findings yield two key insights: (1) Comparing Exp. 1 with Exp. 2, we observe that while 2to3-VAE demonstrates superior reconstruction performance (see table 1), the DiT architecture and absolute position embeddings adopted from Xiang et al. (2025) fail to capture such fine-grained texture features—underscoring the necessity of a synergistic design for diffusion architectures. (2) Our proposed SMDiT and MARoPE each contribute to performance improvements: SMDiT mitigates challenges associated with feature alignment, while MARoPE further boosts performance. Exp. 4, which integrates all the aforementioned proposed components, thereby achieves SOTA results on the evaluated benchmarks by leveraging the combined strengths of these components.

## 5 CONCLUSION

In this paper, we demonstrate that leveraging 2D diffusion-based representations substantially enhances the reconstruction performance of 3D Variational Autoencoders (VAEs). However, we also identify a critical challenge: such fine-grained texture features pose significant challenges to standard DiTs, thereby limiting their ability to capture high-fidelity texture details in 3D generation tasks. We further show that our proposed SMDiT and MARoPE architectures effectively mitigate this limitation. Specifically, these components consistently boost the model's capacity to learn and model detailed texture features, which in turn translates to improved overall 3D generation performance. In particular, our framework achieves markedly superior texture quality for 3D assets with sharp structural details and intricate texture patterns—an area where previous DiT architectures often suffer from notable limitations.

## ETHICS STATEMENT

This work adheres to the ICLR Code of Ethics. In this study, no human subjects or animal experimentation was involved. All datasets used, were sourced in compliance with relevant usage guidelines, ensuring no violation of privacy. We have taken care to avoid any biases or discriminatory outcomes in our research process. No personally identifiable information was used, and no experiments were conducted that could raise privacy or security concerns. We are committed to maintaining transparency and integrity throughout the research process.

## REPRODUCIBILITY STATEMENT

We have made every effort to ensure that the results presented in this paper are reproducible. All code and datasets have been made publicly available in an anonymous repository to facilitate replication and verification. The experimental setup, including training steps, model configurations, and hardware details, is described in detail in the paper. We have also provided a full description of Diff2to3 framwork, to assist others in reproducing our experiments. Additionally, all datasets used

in the paper, including ABO, HSSD, 3D-FUTURE, Objaver-XL, Toys4k, are publicly available, ensuring consistent and reproducible evaluation results. We believe these measures will enable other researchers to reproduce our work and further advance the field.

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

## A    RELATED WORKS

**3D Generation from 2D Guidance.** The breakthrough of high-fidelity 2D generative models (e.g., Stable Diffusion Rombach et al. (2022)) has catalyzed extensive research on leveraging their pre-learned priors to facilitate 3D generative tasks. A core insight of these methods is to distill the rich geometric layout and photorealistic appearance knowledge encoded in 2D diffusion models—trained on massive natural image datasets—into structured 3D representations. DreamFusion Poole et al. (2022) pioneered this direction by introducing Score-Distillation-Sampling (SDS), a key technique that aligns the rendering of a 3D representation (e.g., Neural Radiance Field, NeRF) with the score distribution of a 2D diffusion model, effectively transferring 2D visual priors to 3D space.

Subsequent works have extended the SDS-based pipeline to specialize in image-to-3D generation. Make-it-3D Tang et al. (2023) enhances semantic alignment by first using a Vision-Language Model (VLM) to generate detailed captions for the input reference image, then optimizing a NeRF via a combination of SDS loss (for visual fidelity) and CLIP loss (for semantic consistency with both image and caption). Zero-1-to-3 Liu et al. (2023b) and CAT3D Gao et al. (2024) further refine this paradigm by prioritizing novel view synthesis: they first learn to predict consistent multi-view images from the input single view using 2D diffusion priors, then use these synthesized views to guide the optimization of 3D assets (e.g., NeRF or 3D Gaussian Splatting). One-2-3-45 Liu et al. (2023a) improves upon view consistency and reconstruction efficiency by incorporating geometric constraints (e.g., depth hints) during the multi-view synthesis stage.

Despite these advancements, 2D-guided approaches remain inherently constrained relative to native 3D models. First, the SDS-based optimization process is computationally prohibitive, often requiring hours of iterative refinement to converge to a coherent 3D asset—far from the real-time or near-real-time demands of practical applications. Second, multi-view image generation introduces intrinsic consistency artifacts: 2D diffusion models lack explicit 3D geometric awareness, leading to conflicting visual cues across synthesized views (e.g., mismatched object contours or inconsistent surface textures), which propagate to the final 3D asset and degrade its fidelity.

**Deterministic 3D Reconstruction Models.** A parallel line of research focuses on *native 3D models* designed specifically for image-to-3D tasks, with one prominent branch being large-scale deterministic reconstruction models. These methods aim to reconstruct 3D assets in an end-to-end feed-forward manner, eliminating the need for lengthy iterative optimization. LRM Hong et al. (2023) laid the foundation for this direction by proposing the first transformer-based pipeline tailored for 3D reconstruction: it encodes input 2D images into image tokens, translates these tokens into implicit 3D triplane features via a cross-modal transformer, and finally decodes the triplanes into a NeRF representation. This design unleashes the scalability of large-scale datasets and model parameters, enabling robust single-view to 3D translation. Extending beyond single-view inputs, many successors have integrated multi-view image cues and adopted more expressive 3D output representations (e.g., 3D Gaussian Splatting, 3DGS). GeoLRM Zhang et al. (2024a) introduces a 3D-aware transformer architecture equipped with deformable cross-view attention, which directly aggregates features from multiple input views onto 3D spatial points, enhancing geometric accuracy. GRM Xu et al. (2024b) builds a sparse-view reconstructor that estimates 3D scenes using pixel-aligned Gaussians, leveraging explicit 2D-3D correspondence to preserve fine-grained details. LGM Tang et al. (2024) employs an asymmetric UNet to predict initial 3D Gaussians from multi-view inputs and iteratively fuses them into a coherent 3D asset, balancing efficiency and fidelity. However, the image-to-3D task is fundamentally ill-posed: a single 2D image inherently lacks complete depth and viewpoint information, leading to ambiguous 3D interpretations. Deterministic reconstruction models are inherently unable to model this uncertainty—they map each input image to a single fixed 3D output, rather than a distribution of plausible assets. This limitation often results in unsatisfactory predictions, such as blurry textures, distorted geometry for novel views, or inconsistent details in occluded/unseen regions of the scene, particularly when the input image contains sparse visual cues.

**Stochastic 3D Generation Models.** To address the ambiguity and uncertainty inherent in image-to-3D translation, another branch of native 3D models adopts a *stochastic generation paradigm* based on latent diffusion models (LDMs). These methods model the distribution of plausible 3D assets conditioned on the input image, enabling the synthesis of realistic, diverse, and view-consistent 3D content by leveraging learned data distributions.

**Input Image**

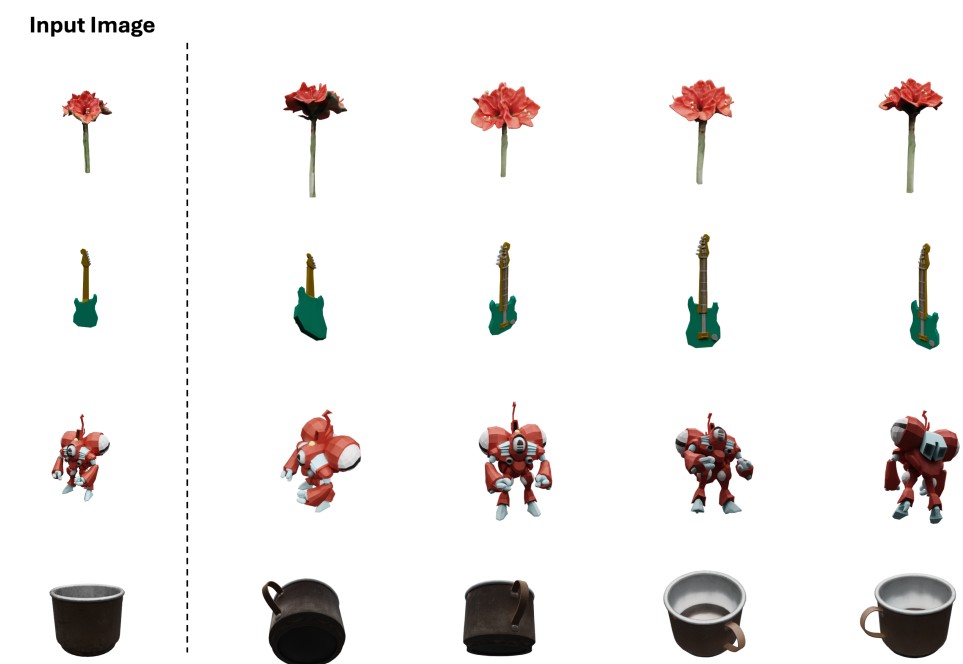

Figure 5: Additional qualitative results

Stochastic 3D generation models are typically categorized by their choice of latent 3D representation, with two dominant families: *vecset-based* and *voxel-based*. Vecset-based methods encode 3D shapes using sets of latent vectors (implicit representations) that can be decoded into neural signed distance functions (SDFs) or occupancy fields. 3DShape2VecSet Zhang et al. (2023) pioneered this representation, demonstrating its ability to capture complex 3D shape variations. Building on this foundation, CLAY Zhang et al. (2024b), TripoSG Li et al. (2025b), and CraftsMan Li et al. (2025a) introduce refinements such as hierarchical vecset encoding and diffusion-based refinement, enabling the generation of highly accurate and detailed 3D shapes.

On the other hand, voxel-based methods leverage explicit spatial structures in the latent space, with recent advances focusing on *sparse 3D structures* to balance expressiveness and efficiency. For example, XCube Ren et al. (2024) and TRELLIS Xiang et al. (2025) embed sparse voxel grids into the latent diffusion framework. This design facilitates efficient feature aggregation within local 3D neighborhoods, reducing computational overhead compared to dense voxel representations while retaining geometric awareness. Subsequent studies like Sparc3D Li et al. (2025c), Ultra3D Chen et al. (2025a) demonstrate the effectiveness of this representation at modeling fine-grained 3D geometry. Despite these strengths, existing sparse voxel-based models often struggle to preserve fine-grained textural details from the input image, due to challenges in aligning sparse 3D latent features with dense 2D image pixels—a gap that motivates further research.

## B ADDITIONAL QUALITATIVE RESULTS

## C LLM USAGE

Large Language Models (LLMs) were used to aid in the writing and polishing of the manuscript. Specifically, we used an LLM to assist in refining the language, improving readability, and ensuring clarity in various sections of the paper. The model helped with tasks such as sentence rephrasing, grammar checking, and enhancing the overall flow of the text.

It is important to note that the LLM was not involved in the ideation, research methodology, or experimental design. All research concepts, ideas, and analyses were developed and conducted by

the authors. The contributions of the LLM were solely focused on improving the linguistic quality of the paper, with no involvement in the scientific content or data analysis.

The authors take full responsibility for the content of the manuscript, including any text generated or polished by the LLM. We have ensured that the LLM-generated text adheres to ethical guidelines and does not contribute to plagiarism or scientific misconduct.

