# OpenReview forum: "Unleashing the Power of 2D Diffusion Representation for High Fidelity 3D Generation"
_ICLR.cc/2026/Conference — ICLR 2026 Conference Withdrawn Submission_

### Official Review · Reviewer_PcCG · 2025-10-28

**Soundness:** 1
**Presentation:** 3
**Contribution:** 2
**Rating:** 4
**Confidence:** 5

**Summary:**

This paper introduces Diff2to3, a novel framework for high-fidelity, joint geometry and texture generation for 3D content from a single 2D image. The authors identify two primary weaknesses in existing joint-generation models: (1) the use of coarse-grained semantic features (like DINOv2) for latent representation, which loses fine texture details during encoding , and (2) ineffective cross-modal alignment between 2D image features and 3D structural latents, which hinders the accurate mapping of textures.

This paper claim three core contributions:

**2to3-VAE with FLUX Features:** The paper replaces the commonly used DINOv2 features with **FLUX diffusion features** as input to their VAE encoder. The rationale is that FLUX features are optimized for reconstruction and preserve low-level visual details. They also highlight a significantly more favorable compression ratio (2:1) compared to DINOv2 (128:1), which minimizes information loss.

**SMDiT (Sparse-structure Multi-modal Diffusion Transformer):** A novel diffusion transformer architecture designed to handle sparse 3D latents. It uses a **double-stream** mechanism to process 3D sparse tokens and 2D image tokens separately before fusing them with **joint attention**. This is followed by **single-stream** blocks to enhance cross-modal interaction

**MAROPE (Modal-Aware Rotary Position Embedding):** A clever and practical positional embedding scheme to implicitly align the 2D and 3D modalities without requiring camera parameters. It projects 2D image patches onto a **"virtual plane"** in 3D space.

**Strengths:**

1. Paper's insightful diagnosis of a core limitation in existing joint generation frameworks: the loss of fine-grained texture details. The authors compellingly argue this stems from the use of high-compression, semantic-focused features like DINOv2 (a 128:1 ratio), which discard critical low-level information during encoding. The proposed solution—leveraging reconstructive, low-compression (2:1) FLUX diffusion features for the VAE input—is well-motivated and directly addresses this bottleneck, enabling the latent space to preserve the fine-grained visual details necessary for high-fidelity 3D generation.

2. Novel generator architecture, which is co-designed to model the high-fidelity latent space produced by the FLUX-based VAE. It introduces the SMDiT (Sparse-structure Multi-modal Diffusion Transformer), which effectively fuses information from the 2D image condition and the 3D sparse voxel latents through a principled dual-stream and joint-attention mechanism. The proposed MAROPE (Modal-Aware Rotary Position Embedding) offers an elegant and practical solution for 2D-3D spatial alignment. By mapping 2D patches to a "virtual plane" , it allows the model to learn implicit correspondences without requiring explicit camera parameters, overcoming a significant limitation of prior 3D-aware embedding methods.

3. In the experimental sections, Diff2to3 consistently achieves state-of-the-art performance across both reconstruction and generation benchmarks, outperforming prior methods such as TRELLIS, GaussianAnything, and InstantMesh on standard metrics including CLIP, FD, and KD.

**Weaknesses:**

1. The author claimed that "When compressed to the target 8-dimensional latents, this dimensionality reduction (16→8) results in a modest compression ratio of 2:1—far lower than DINOv2’s 128:1. This reduced compression ratio minimizes information loss, enabling the retention of finegrained textural details in the latent space."

   There is a  significant contradiction that emerges when comparing the reconstruction results in Table 1, which show a slight advantage for the 2to3-VAE, with the generative ablation study in Table 3. Specifically, Exp 2 in Table 3 demonstrates that isolating the new 2to3-VAE and pairing it with the baseline generator causes a substantial degradation in performance (e.g., CLIP score drops from 98.02 to 96.96) compared to the Exp 1 baseline, undermining the claim of the VAE's independent superiority.

   The paper's explanation—that this is due to the baseline DiT's inability to "capture such fine-grained texture features" is not fully convincing. This result suggests that the FLUX VAE's benefit is not independent. Instead, it seems the new VAE and the SMDiT generator are highly co-dependent, and the VAE may not offer a general advantage without the specific SMDiT architecture designed to handle its outputs.

2. The ablation study in Table 3 is incomplete. It does not properly separate the effect of the new VAE from the new SMDiT generator. The table shows the baseline (Exp 1: DINOv2-VAE + DiT) and the full model (Exp 4: FLUX-VAE + SMDiT). However, some experiments are missing, such as testing the original DINOv2-VAE with the new SMDiT/MAROPE architecture.  Please add more details to the ablation study.

3. Unfair Quantitative Comparison: The primary quantitative comparison in Table 2 is not conducted under a perfectly fair setting, which complicates the interpretation of the results.The proposed model ("Ours") has 821M parameters, which is larger than the key baseline, TRELLIS (which has 770M parameters).While the proposed model achieves the best scores, the margin of improvement on several metrics is not substantial given its larger size. For example, the CLIP score improves only slightly (98.45 vs. 98.02), as do the incep metrics. It is unclear how much of the performance gain is due to the novel architecture versus the simple increase in model capacity. Meanwhile, competing methods like InstantMesh generate meshes, while this work outputs 3D Gaussian splatting results. Gaussian renderers usually produce smoother and more photorealistic images than mesh renderers, which naturally leads to higher CLIP, FD, and KD scores. Therefore, the comparison setting is biased. To ensure fairness, the authors should replace the Gaussian decoder with a mesh decoder (e.g., from TRELLIS) and compare results in mesh form. This experiment would be crucial to confirm the claimed superiority.

4. The paper clearly states that *“these methods typically incur computational costs on the order of minutes when generating both geometry and texture”*, identifying efficiency as a main motivation. However, the proposed framework still builds on the heavy Flux-based DiT backbone, and no inference-time or throughput comparison on the same hardware is reported. To validate this motivation, quantitative results such as per-sample latency or overall efficiency on matched machines should be provided.

5. The paper claims to improve fine-grained textures, but the visual evidence provided in Figure 4 and Figure 5 is limited. A more extensive set of qualitative comparisons is needed to fully substantiate the robustness of this claim.

**Questions:**

Please refer to Weakness

---

### Official Review · Reviewer_mdqP · 2025-10-29

**Soundness:** 2
**Presentation:** 3
**Contribution:** 2
**Rating:** 4
**Confidence:** 4

**Summary:**

This paper identifies a key limitation in state-of-the-art (SOTA) 3D generation methods: the loss of fine-grained texture details during latent feature learning. To address this, the authors propose Diff2to3, a joint geometry and texture generation framework. The primary contributions are threefold: 1) 2to3-VAE, a variational autoencoder that modifies the SOTA TRELLIS VAE by using features from a 2D diffusion model instead of DINOv2, aiming to better preserve low-level texture details. 2) SMDiT, a novel Sparse-structure Multi-modal Diffusion Transformer that uses double-stream and single-stream blocks to process 3D sparse voxel tokens and 2D image tokens for improved alignment. 3) MAROPE, a new Modal-Aware Rotary Position Embedding designed to learn implicit 2D-3D correspondences. The authors claim that this combined approach outperforms existing SOTA methods, particularly in generating 3D assets with high-fidelity, image-consistent textures.

**Strengths:**

1. Important Problem: The paper tackles a critical and timely challenge in 3D generation: achieving high-fidelity texture preservation from input 2D images. This is a significant bottleneck for current methods.
2. Logical Intuition for VAE: The core idea of the 2to3-VAE—using features from a reconstruction-focused model (FLUX) rather than a semantic-focused model (DINOv2) to build the latent space—is intuitive for a task that requires preserving fine-grained details.
3. Strong Reconstruction Results: This intuition is validated by the reconstruction experiments in Table 1, where the proposed 2to3-VAE significantly outperforms the TRELLIS VAE on all reconstruction metrics (SSIM, PSNR, LPIPS).
4. Effective Position Embedding: The proposed MAROPE module appears to be an effective contribution, as the ablation study suggests it is responsible for the final performance gain.

**Weaknesses:**

1. (Major Flaw) Ablation Study Contradicts Thesis: The paper's central narrative is that using 2D diffusion representation (via 2to3-VAE) and a new multi-modal transformer (SMDiT) improves generation. However, the ablation study in Table 3 shows the exact opposite.
o Exp 1 (TRELLIS baseline) has an $FD_{dinov2}$ of 146.14.
o Exp 2 (substituting 2to3-VAE) degrades performance significantly to 210.50.
o Exp 3 (adding SMDiT) is still worse than the baseline at 159.34.
o Only in Exp 4, with the addition of the MAROPE module, does the performance finally surpass the baseline (122.90). This strongly suggests that the paper's two main contributions (2to3-VAE and SMDiT) are ineffective or even detrimental for the generation task, and the entire performance gain comes from the MAROPE module, which is presented as a secondary contribution. This is a severe disconnect between the paper's claims and its evidence.
2. Limited Novelty: The core contributions feel incremental. The 2to3-VAE is a simple feature-swapping (FLUX for DINOv2) on the existing TRELLIS VAE architecture. The SMDiT's dual-stream/single-stream design is heavily inspired by prior work in 2D diffusion transformers.
3. Insufficient Qualitative Evaluation: For a paper claiming to solve high-fidelity texture generation, the qualitative results are sparse. Figure 4 and the appendix figure show only a handful of static, multi-view images. This is not sufficient to evaluate 3D consistency, geometric quality, or texture fidelity from arbitrary novel viewpoints.
4. Lack of Video/Interactive Demonstration: The submission includes no supplementary website with video renders or interactive 3D models. This has become a standard and essential component for evaluating 3D generation papers, as static images can be easily "cherry-picked." Without videos, it is impossible to verify the paper's claims about 3D quality.

**Questions:**

1. The primary question: Why does the 2to3-VAE, which demonstrates superior reconstruction quality (Table 1), result in significantly worse generation performance (Table 3, Exp 2 vs. Exp 1)? Please explain this disconnect, as it seems to invalidate the paper's core hypothesis.
2. Have the authors run the most critical missing ablation: applying the MAROPE module directly to the TRELLIS baseline (i.e., DINOv2 VAE + DiT + MAROPE)? This is essential to isolate MAROPE's contribution and confirm if 2to3-VAE and SMDiT are truly unnecessary or harmful.
3. Given the claims of high-fidelity 3D generation, can the authors please provide a supplementary website with video renders (not just static images) for the objects shown in Figure 4 and compared against the baselines? This is necessary to properly assess 3D consistency and texture quality.

---

### Official Review · Reviewer_gnaG · 2025-11-01

**Soundness:** 3
**Presentation:** 3
**Contribution:** 2
**Rating:** 4
**Confidence:** 4

**Summary:**

The paper proposes Diff2to3, a joint image-to-3D generation framework that aims to preserve fine-grained textures by (i) replacing the DINOv2 features used in TRELLIS’s VAE with FLUX (2D diffusion) features for constructing the latent input volume, and (ii) introducing a Sparse-structure Multi-modal Diffusion Transformer (SMDiT) together with a Modal-Aware Rotary Position Embedding (MARoPE) to improve 2D–3D alignment.

**Strengths:**

1. Compelling motivation and observation. It’s true that DINO-style features struggle with fine textures; moving to FLUX features is a clear, plausible improvement path.

2. The paper provides a clean ablation isolating VAE feature choice, SMDiT, and MARoPE; metrics consistently trend in the right direction.

3. The paper is well structured and easy to follow.

**Weaknesses:**

1. Limited technical contribution. Beyond replacing DINO with FLUX in the TRELLIS-style VAE, the architectural differences are modest.

2. MARoPE is minor. MARoPE effectively adds a depth “plane” index (u, v, z_{max}+1) for image tokens—i.e., one extra modality-aware dimension—rather than a principled new 2D–3D correspondence mechanism. The gain shown is small; theoretical or empirical analysis is thin.

3. Claims vs. evidence gap for “fine-grained texture.” Qualitative and quantitative results are mostly on simple objects (smooth geometry, simple materials). The paper claims fine-grained fidelity but does not convincingly stress-test intricate textures (fabrics, wood grain, brushed metal, text/logos under oblique views, specular/anisotropic surfaces, etc.). I would like to raise my score if the authors could prove that Diff2to3 can generate significantly better textures than the state-of-the-art methods for complex objects.

**Questions:**

I agree that DINO features lack sufficient texture information. However, alternative features from image diffusion VAEs—such as those from Stable Diffusion VAE—do contain rich information and can be reconstructed into the original image using a decoder. I wonder whether these other VAE-encoded features perform as effectively as FLUX features.

---

### Official Review · Reviewer_jHnC · 2025-11-02

**Soundness:** 1
**Presentation:** 2
**Contribution:** 1
**Rating:** 0
**Confidence:** 4

**Summary:**

The paper addresses the problem of high-fidelity 3D generation from single images, aiming to jointly model both geometry and appearance. Prior approaches typically decouple shape and texture generation, which often leads to shape–texture misalignment, or they rely on visual features (e.g., DINOv2) that capture coarse semantics but lack fine-grained texture detail.

To overcome these limitations, the proposed method, Diff2to3, integrates 2D diffusion model features from large-scale pretrained image models to construct 3D structured latents suitable for geometry and texture generation, building on the sparse voxel representation of TRELLIS. The framework further introduces a Sparse-structure Multi-modal Diffusion Transformer (SMDiT) that effectively fuses information between 2D image tokens and 3D latent voxels, and a Modal-Aware Rotary Position Embedding (MARoPE) that facilitates implicit cross-modal alignment between 2D and 3D representations without requiring explicit camera calibration.

The method is evaluated on the Toys4K dataset for both 3D reconstruction and image-conditioned 3D generation, showing consistent improvements over strong baselines.

**Strengths:**

The paper addresses a challenging and timely problem in 3D generation, the joint modeling of geometry and texture from single images while maintaining both structural accuracy and high-fidelity appearance. Experimental results demonstrate clear and consistent improvements over recent SOTA methods in both 3D reconstruction and image-conditioned generation.

**Weaknesses:**

- Several methodological aspects lack clarity and depth. First, the description of the SMDiT architecture is incomplete. It is unclear how the sparse structure and corresponding latent features are generated. In TRELLIS, this is a two-stage process (structure and latent generation, both conditioned on the input image/text prompt), whereas in Diff2to3 only the second stage appears to be discussed. The paper does not clearly specify whether the sparse structure is precomputed or learned jointly, and the proposed modification, replacing cross-attention with a joint attention mechanism, appears incremental. Second, the explanation of the MARoPE module is vague. The paper only provides a conceptual description of mapping 2D tokens to a “virtual plane” in a shared 3D coordinate system, without a precise mathematical formulation of the positional encoding or its integration into attention layers. A comparison with prior 3D-aware rotary embeddings such as [1]  is notably missing.
- The claim that diffusion-based visual features (FLUX) outperform DINOv2 features is not well supported by the experiments. As shown in Table 3, using the TRELLIS VAE (with DINOv2) actually performs the proposed VAE variant (row 2), contradicting the claim that diffusion features preserve more fine-grained details (lines 246–249). This weakens the stated motivation for replacing DINO features.
- Key ablations are missing or incomplete. In particular, the paper should evaluate the MARoPE module compared to prior RoPE variants (e.g., Romantex) and the effect of using MARoPE and the 2to3-VAE independently.
- The method provides only incremental improvements over TRELLIS, with limited architectural novelty, and the VAE training setup lacks sufficient detail (e.g., loss term definitions).

[1] Feng, Yifei, et al., "Romantex: Decoupling 3d-aware rotary positional embedded multi-attention network for texture synthesis." Proceedings of the IEEE/CVF International Conference on Computer Vision. 2025.

**Questions:**

- Is there a specific reason why Diff2to3 is conditioned solely on images, rather than incorporating text prompts as additional conditioning signals (as in TRELLIS or related multimodal frameworks)?
- Why do the evaluations focus exclusively on appearance fidelity (texture quality), without reporting metrics or visual analyses that assess the geometric accuracy of the generated 3D assets?
- Sentences in lines 267-268 and lines 268-269 are duplicates.

---

### Note · Authors · 2025-11-13

**Comment:**

Thank all the reviewers for the insightful comments. We'll further improve our paper based on your constructive suggestions.

**Withdrawal Confirmation:**

I have read and agree with the venue's withdrawal policy on behalf of myself and my co-authors.